# Head Injury Evaluation and Ambulance Diagnosis (HOME) Study protocol: a feasibility study assessing the implementation of the Canadian CT Head Rule in the prehospital setting

Naif Alqurashi ,[1,2] Steve Bell ,[3] Simon D Carley,[4] Fiona Lecky ,[5] Richard Body[1,4]

For numbered affiliations see end of article.

**Correspondence to**
Naif Alqurashi;
nalqurashi1@ksu.edu.sa

## ABSTRACT

**Introduction** Traumatic brain injury (TBI) is a common presentation in the prehospital environment. At present, paramedics do not routinely use tools to identify low-risk patients who could be left at scene or taken to a local hospital rather than a major trauma centre. The Canadian CT Head Rule (CCHR) was developed to guide the use of CT imaging in hospital. It has not been evaluated in the prehospital setting. We aim to address this gap by evaluating the feasibility and acceptability of implementing the CCHR to patients and paramedics, and the feasibility of conducting a full-scale clinical trial of its use.

**Methods and analysis** We will recruit adult patients who are being transported to an emergency department (ED) by ambulance after suffering a mild TBI. Paramedics will prospectively collect data for the CCHR. All patients will be transported to the ED, where deferred consent will be taken and the treating clinician will reassess the CCHR, blinded to paramedic interpretation. The primary clinical outcome will be neurosurgically significant TBI. Feasibility outcomes include recruitment and attrition rates. We will assess acceptability of the CCHR to paramedics using the Ottawa Acceptability of Decision Rules Instrument. Interobserver reliability of the CCHR will be assessed between paramedics and the treating clinician in the ED. Participating paramedics and patients will be invited to participate in semistructured interviews to explore the acceptability of trial processes and facilitators and barriers to the use of the CCHR in practice. Data will be analysed thematically. We anticipate recruiting approximately 100 patients over 6 months.

**Ethics and dissemination** This study was approved by the Health Research Authority and the Research Ethics Committee (REC reference: 22/NW/0358). The results will be published in a peer-reviewed journal, presented at conferences and will be incorporated into a doctoral thesis.

**Trial registration number** ISRCTN92566288.

## INTRODUCTION

Traumatic brain injury (TBI) is widely recognised as being one of the most common causes of death and disability especially

### STRENGTHS AND LIMITATIONS OF THIS STUDY

⇒ To our knowledge, this study will be the first to evaluate the feasibility and acceptability of implementing the Canadian CT Head Rule (CCHR) in the prehospital setting, filling an important gap in the existing literature.

⇒ The study has a limited sample size, as its primary objective is to assess the acceptability and feasibility of implementing the CCHR in the prehospital setting.

⇒ The findings of this study will be used in the future to inform the design of a fully powered diagnostic test accuracy study.

among young adults[1]; however, it is becoming more frequent in the older population.[2] TBI is a heterogeneous pathophysiological condition that varies widely in terms of its aetiology, severity, clinical presentation and outcomes. This broad spectrum of severity creates clinical challenges, such as the inappropriate allocation of major trauma centre (MTC) resources to less serious cases.[3]

The goal of prehospital trauma triage tools is to efficiently and promptly direct patients to the medical facility best suited to their needs and the available resources. Most TBIs are mild in nature and can be treated at a local trauma unit or non-trauma centres.[4] There is no evidence indicating that these patients should be transported to a specialist neurological centre. Therefore, accurate prehospital triage of these patients will reduce the likelihood of MTCs being overwhelmed with patients suffering from mild TBI or non-TBI injuries as well as reducing the costs that might result from overtriaging patients who are unlikely to benefit from the costly resources of an MTC.

The cranial CT scan is considered the gold standard for the emergency evaluation of patients with suspected intracranial injury.[5] Various validated clinical decision tools, including the Canadian CT Head Rule (CCHR), have been developed through extensive prospective studies to assess the need for a CT scan after a minor head injury.[6 7] Comparative studies have examined various clinical tools, most notably the New Orleans Criteria and the CCHR, revealing that although both demonstrate similar sensitivity, the CCHR generally provides greater specificity.[8–10] Several studies have shown that the CCHR achieves 100% sensitivity in identifying neurosurgical needs and captures significant CT findings, offering greater specificity (36–65%) compared with other decision rules.[9–11]

The National Institute for Health and Care Excellence (NICE) has guidelines that largely follow the CCHR to determine when to perform CT scans in cases of head injury.[12] This approach is supported by evidence showing that the CCHR rule is the most cost-effective among various decision rules evaluated.[13] Although the number of CT scans has increased, the admission rate has been halved (from 9% to 4%), resulting in a saving of £3381 per 100 patients.[14 15] The implementation of NICE head injury guidelines indicates that the higher costs of CT scans, averaging about £100 per scan, are balanced by the savings from reduced hospital admissions, estimated to be around £847 per patient.[13]

Considering that paramedics have access to all the necessary data for the CCHR, it might be feasible to use the CCHR in the prehospital setting to promote consistency in care and optimise resource utilisation. Prehospital implementation of the CCHR could enable paramedics to transport patients to lower-level or non-trauma centres, or even (if shown to be sufficiently sensitive) safely leave patients at the scene. However, no studies have explored whether clinical decision aids can be used to rule out neurosurgically significant TBI in patients with apparently mild TBI in the prehospital field.

In England and Wales, around 1 million emergency department (ED) visits annually are for head injuries, with 90% being minor (Glasgow Coma Scale (GCS) 13–15), and the majority of patients are transported by ambulance services.[16] Optimising healthcare resource use and enhancing patient care experiences can be greatly advanced by reducing unnecessary ambulance transports to distant MTCs, when appropriate and feasible.[17 18] Integrating the CCHR into prehospital field presents a valuable opportunity to improve the triage process, allowing prehospital care providers to make informed decisions regarding the necessity of hospital transport and the level of care required, thereby optimising healthcare resources and potentially reducing healthcare costs. This approach also opens the door to exploring alternative care pathways for such patients, ensuring they receive appropriate and timely care without overwhelming hospital resources.

To obtain reliable estimates of the diagnostic accuracy of the CCHR, a large, multicentre study is warranted. This would be expensive and time-consuming; however, the findings of such a study could inform changes to clinical practice. Successful delivery of such research would depend heavily on adherence to the study protocol and the rates of recruitment attrition. Before embarking on such a study, it is necessary to evaluate the viability of the intended study processes.[19]

The purpose of this study is to establish the feasibility of a fully powered diagnostic test accuracy study to evaluate the accuracy of the CCHR when applied by paramedics to patients with mild TBI in the prehospital setting.

## Research question

Is it feasible to conduct a prospective clinical study to evaluate the diagnostic accuracy of the CCHR in the prehospital field to assist with making informed decisions regarding the appropriate level of care for patients with minor head injuries?

## Objectives

► To determine the feasibility of a study to evaluate the diagnostic accuracy of the CCHR in the prehospital environment.
► To evaluate the interobserver reliability of the CCHR between prehospital clinicians and ED physicians.
► To evaluate the clinical sensibility of using the CCHR in the prehospital environment and to measure paramedics' attitudes toward the CCHR.
► To obtain an estimate of the diagnostic accuracy of the CCHR when used in the prehospital setting.

## METHODS
### Study design and setting

A prospective, multicentre feasibility study will be conducted in Greater Manchester, a region within the service area of the North West Ambulance Service (NWAS). NWAS is a National Health Service (NHS) trust that provides emergency medical services and patient transportation across the North West region of England. It covers an extensive area of over 5400 square miles, providing emergency and non-emergency medical services to a population of over 7.5 million people. There are two adult MTCs in Greater Manchester, which are part of the network covered by NWAS. These are Salford Royal Hospital and Manchester Royal Infirmary. In addition to these MTCs, the network also includes several trauma units and local emergency hospitals, each playing a distinct role in the trauma care pathway. The study protocol has been prospectively registered through the ISRCTN registry (study ID: ISRCTN92566288).

For the purpose of the study, we modified the CCHR for prehospital use (table 1). The original rule excluded patients <16 years, patients with a bleeding disorder or users of oral anticoagulants and/or antiplatelet and or those experiencing a post-traumatic

**Table 1** The original and modified versions of the Canadian CT Head Rule

| Canadian CT Head Rule (original) | Canadian CT Head Rule (modified) |
|---|---|
| **High risk (for neurosurgical interventions)** | **High risk (for neurosurgical interventions)** |
| GCS score <15 at 2 hours after injury | GCS score <15 at presentation |
| Suspected open or depressed skull fracture | Suspected open or depressed skull fracture |
| Any sign of basal skull fracture | Any sign of basal skull fracture |
| Vomiting ≥2 episodes | Vomiting ≥2 episodes |
| Age ≥65 years | Age ≥65 years |
| **Medium risk (for brain injury on CT)** | **Medium risk (for brain injury on CT)** |
| Amnesia before impact >30 min | Amnesia before impact >30 min |
| Dangerous mechanism of injury (pedestrian struck by vehicle, ejection from vehicle, fall from >3 ft or >5 stairs | Dangerous mechanism of injury (pedestrian struck by vehicle, ejection from vehicle, fall from >3 ft or >5 stairs |

Patients with any positive high-risk or medium-risk criteria will be considered to have a 'positive' outcome (requiring transport to a hospital).
GCS, Glasgow Coma Scale.

seizure. In the modified version, only patients under the age of 16 years will be excluded. However, patients taking anticoagulant or antiplatelet drugs will be included and will be considered 'positive' with the modified CCHR (ie, would require transport to hospital). Anticoagulant use is defined as the use of warfarin, edoxaban, dabigatran, apixaban and rivaroxaban. Antiplatelets use is defined as the use of clopidogrel, ticlopidine, ticagrelor, dipyridamole and prasugrel. Figure 1 displays a visual representation of the recruitment process. At the time of writing in April 2024, the study is actively in the recruitment phase and open for participant enrolment.

## Participants
### Patient eligibility
#### Inclusion criteria
This study will be conducted using a convenience sample of adult patients with mild TBI who will be transported to Northern Care Alliance NHS Foundation Trust and Manchester University Hospital NHS Foundation Trust. Only those patients who meet the following criteria will be included:
► Adult patients (≥16 years) who receive an emergency ambulance response for a primary complaint of head injury. This includes those involved in motor vehicle crashes (MVCs), provided they meet the other inclusion criteria.

► GCS score of 13–15 at the time of assessment by attending paramedics.
► Patients transported to the hospital for clinical care.

#### Exclusion criteria
► Secondary transfers of patients including interfacility transport.
► Penetrating skull injury.
► Trauma to other body regions that require clinical treatment, indicating multisystem trauma. This includes patients involved in MVCs where the treating paramedic assesses the presence of injuries beyond isolated head trauma that require clinical intervention. Paramedics will apply their clinical judgement to identify and exclude such cases.
► Prisoners.

## Study duration and sample size
This is a feasibility study, in which evaluating recruitment rate is an outcome; therefore, it is not possible to specify a sample size a priori. However, we intend to recruit participants for a fixed period of 6 months. Based on data routinely collected by the Manchester University NHS Foundation Trust, a total of 5332 patients arrived in the ED by ambulance with a primary complaint of head injury over a period of 4 years and 9 months up to November 2020. We intend to recruit participants from two similarly sized MTCs. We anticipate that approximately 1100 patients will be eligible to participate in this study during the recruitment period. Accounting for the availability of trained paramedics on the study delegation log and concurrent injuries to other body systems that would exclude patients from participating in this study, approximately 100 participants can be recruited to this study. This number of participants is generally accepted as being appropriate for studies that focus on assessing feasibility and acceptability.[20]

## Outcomes
We will study the following feasibility outcomes:
► The number of eligible patients who are approached.
► The proportion of those approached who consent to participate in the study.
► The completeness of data collection.
► The completeness of follow-up.
► The number and proportion of patients with neurosurgically significant TBI.
► Determination of the interobserver reliability of the CCHR completed by prehospital care providers and ED physicians.
► Determination of the interobserver reliability for each component of the CCHR assessed by prehospital care providers and ED physicians.
► The diagnostic performance and clinical sensibility of the CCHR in the opinion of attending paramedics.
► Determination of the acceptability of trial processes as perceived by paramedics and patients.

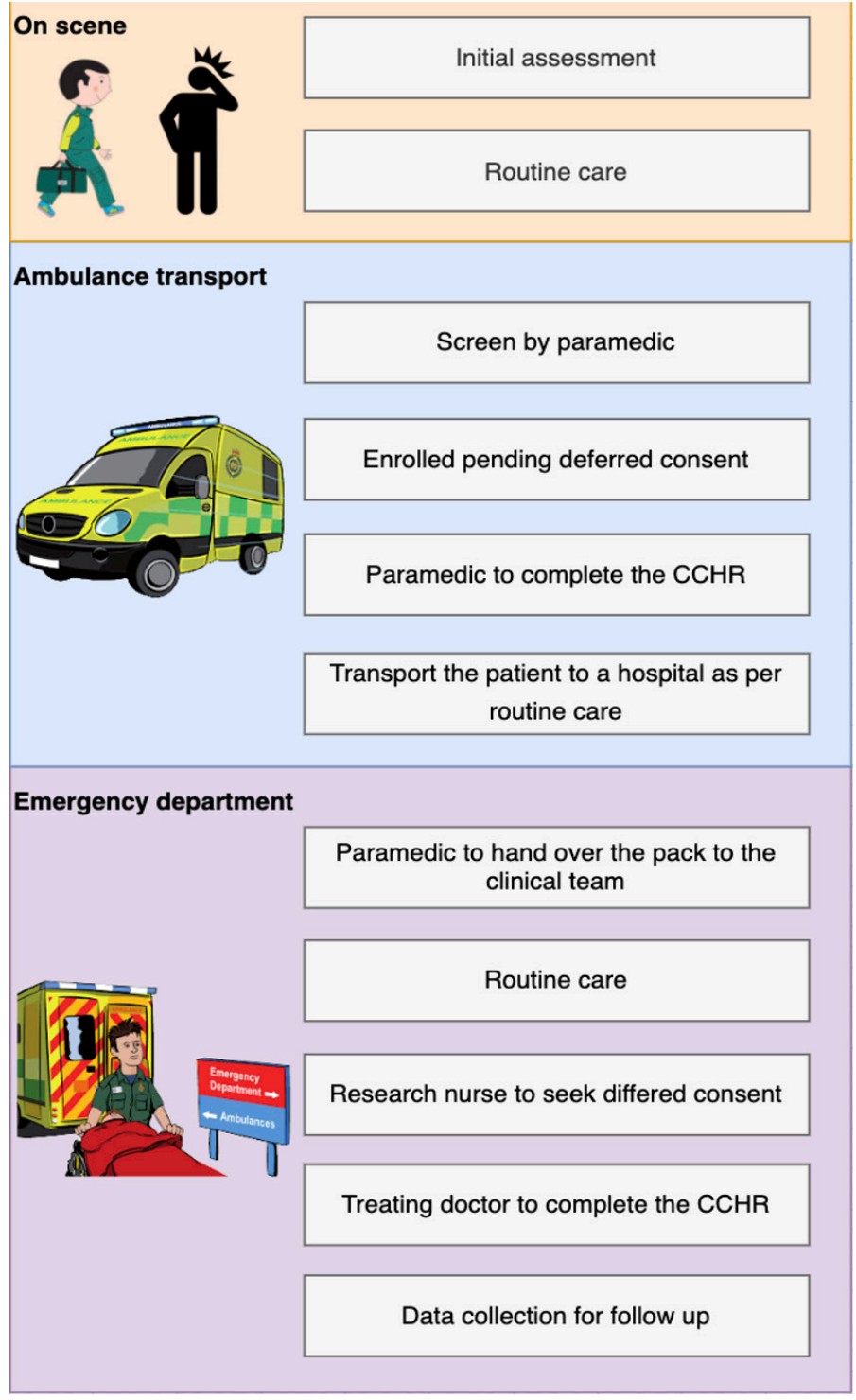

**Figure 1** Study flow chart. CCHR, Canadian CT Head Rule.

### Measuring and defining outcomes

Feasibility outcomes will be assessed using screening logs and case report forms (CRFs) completed by attending paramedics during the course of the study. At the end of the study period, a minimum of two clinicians (affiliated with the relevant NHS trusts) will review in-hospital patient records to diagnose the secondary outcomes of this feasibility study (neurosurgically significant TBI, abnormal CT findings and death within 7 days). A third independent investigator will be invited to adjudicate any discrepancies. Diagnostic adjudication will be blinded to the final interpretation of the CCHR. Additionally, a neuroradiology consultant/expert will be consulted in case of the CT scan diagnosis is inconclusive.

Neurosurgical interventions are defined as any neurosurgical intervention including intracranial pressure monitoring or an Abbreviated Injury Scale (AIS) with the severity ≥3 in the head region. Abnormal head CT

findings will be classified as normal or abnormal and are defined as at least one of the following: skull fractures, intracranial lesions, such as cerebral contusion and traumatic intracranial haemorrhages, or other clinically significant findings requiring hospital admission, neurosurgical interventions and routine follow-up.

## Data collection
### Prehospital setting
During the study period, all patients will receive standard prehospital care in accordance with local guidelines. Prior to their participation in the study, all participating paramedics will be provided with bespoke training specifically designed for the study. This training will cover all aspects of the study protocol, including eligibility assessment, the consent process, data collection procedures and guidance on interpreting the CCHR. To ensure accessibility and flexibility, the training will be offered both face-to-face and through various online sessions conducted via Zoom. Additionally, the training sessions have been recorded and made available online for those unable to attend live. This approach allows paramedics to access the training material at their convenience, ensuring that all participants receive comprehensive training in the study protocols and data collection methods.

Prehospital data will be collected using Research Electronic Data Capture (REDCap), a secure web-based platform hosted at the University of Manchester. REDCap is a software application that was created to facilitate the process of collecting and managing research data. Prior to recruitment, a training session will be held to equip participating sites with data collection skills. In addition, paper CRFs will be available as an alternative method of data collection. It is expected that this approach will provide flexibility and ensure continuity in data collection. A variety of data will be collected in the prehospital setting, including, but not limited to:
► Baseline demographics information (ie, gender–age).
► Anticoagulant use.
► Injury characteristics (ie, mechanism of injury, place).
► Medical history.
► Initial chief complaint (ie, vomiting, headache, loss of consciousness, amnesia).
► Prehospital physiological parameters (ie, GCS, vital signs).
► Clinical criteria required to use the CCHR.
► Destination hospital.

Paramedics will record these data contemporaneously, either prior to or immediately following handover at hospital.

### In-hospital environment
REDCap will be used to collect hospital data, and a paper CRF will also be available as an alternative. Emergency physicians at the receiving hospital will assess the patients to determine whether a CT scan is necessary in accordance with local and national guidelines. Relevant data about imaging studies undertaken and their findings will

be extracted from hospital records. The routine radiology reports will be used to extract the follow-up data required for this study. Where there is uncertainty about the interpretation of the radiology report, a second radiologist will be asked to interpret the images.

Further, emergency physicians at the receiving hospitals will be asked to assess the patients using the CCHR and record their findings, blinded to the assessments made by participating paramedics. These data will be used to determine the interobserver reliability of the CCHR between paramedics in the prehospital environment and clinicians in the ED. Additionally, in-hospital medical records will be reviewed to obtain information regarding neurosurgical interventions undertaken during the patient's hospital admission.

### Follow-up
Following the initial presentation to the ED, patients will be followed up to the time of hospital discharge. Data will be collected from the clinical records of the patient, including, but not limited to:
► GCS on arrival in hospital.
► Details of the clinician's assessment of the CCHR.
► Outcome of any brain imaging (ie, CT and/or MR scans, recorded as part of routine practice).
► AIS in relation to head injury.
► Injury Severity Score.
► Mortality status.
► Neurosurgical procedures and/or surgery.

## Evaluation of acceptability
We will use a qualitative approach to evaluate the acceptability of trial processes used to study participants. We will purposively sample (a) study participants and (b) paramedics. The intention is to maximise heterogeneity (according to age, sex and ethnic origin, and for paramedics, seniority and base ambulance station). Patients and participating paramedics will be interviewed for the purpose of exploring their experiences of participating in a study of this nature. Previous emergency care research has demonstrated the value of this method in providing information regarding the feasibility and design of a larger study.[21]

To maximise convenience for participants and thereby recruitment, participants will be invited to take part in a semistructured interview either by telephone, face-to-face or video conference. Informed consent will be obtained from all participants. We will explore the factors that facilitate or discourage participation in the research. Paramedics and study participants will be offered a £30 high street voucher to thank them for taking part. Telephone and sign language interpreters will be provided where necessary. The interview topic guides (online supplemental appendix 1) consist of a number of questions and prompts concerning the patients and paramedics' experiences of participating in this study, as well as the paramedics' perceptions of the viability of the CCHR. Prior to commencement of the interviews, two pilot interviews will

be carried out. To determine the relevance, clarity and duration required to complete each interview, one will be with a paramedic and the other will be with a patient.[22]

The Ottawa Acceptability of Decision Rules Instrument (OADRI) is a 12-item instrument that will be used to measure paramedics' attitudes toward the CCHR and clinical sensibility.[23] Paramedics will be asked to provide their level of agreement with 12 statements about the CCHR using a 6-point Likert scale. The first seven statements are worded in such a way that a higher number indicates higher acceptance, while the last five are worded in the opposite manner to minimise the acquiescence bias.[24]

## Consent process

An exception to the normal process of informed consent is established for patients in emergency situations who are unable to provide consent. As stated in the Declaration of Helsinki, "for a research subject who is legally incompetent, physically or mentally incapable of giving consent or is a legally incompetent minor, the investigator must obtain informed consent from the legally authorised representative in accordance with applicable law. These groups should not be included in research unless the research is necessary to promote the health of the population represented and this research cannot instead be performed on legally competent persons."[25]

In this study, we will include patients in the prehospital environment who are receiving an emergency ambulance response having sustained a head injury. The nature of this presentation means that the mental capacity to provide informed consent for participation in research will be compromised, especially given the time-critical nature of the emergency medical response. Therefore, we will adopt a deferred consent model, which is widely used in research of this nature.[26] Deferred consent in the emergency setting is a practice where medical treatment or research participation is initiated without prior consent from patients due to urgent circumstances and their compromised mental capacity, with the intention to obtain consent retrospectively once the patient's mental capacity has improved.[27]

Paramedics will enrol eligible participants, enabling them to collect the required data. Patient care will be unchanged. We will then seek deferred written informed consent after the patients have received their initial treatment in hospital, and once they have regained their mental capacity.

When seeking written informed consent, a participant information sheet will be provided. Deferred consent may be obtained by either of the following means: (a) a member of staff may seek written informed consent at the receiving hospital; or (b) deferred consent may be sought in the community by post (written informed consent), email or telephone. While seeking deferred consent, staff will be available to answer any questions that the participants may have.

For participants lacking capacity, we will seek consent from their relative or legally authorised representative, which is known as proxy consent.[28] A research nurse will briefly explain to them that the patient will receive the usual hospital care for TBI. The responsible nurse will also clarify that only basic clinical data will be reported using a CRF to be analysed to improve prehospital care. A brief information sheet will be provided upon request. The ability of participants to provide consent can be assessed at any point during the study period. Participants who regain capacity will be provided with a detailed participant information sheet, and their written consent to continue in this feasibility study will be obtained. If the patient does not regain capacity, we will seek deferred assent from a relative or professional legal representative to retain the patient's data in the study.

## Data analysis

### Quantitative data

A descriptive analysis will be conducted based on the baseline data for patients enrolled in the study to characterise the demographic and clinical profile. This will include the number of eligible participants who were approached, the proportion who consented to participate, data completeness (expressed as number and percentage) for each variable collected and number and percentage of participants completing follow-up. We will also report recruitment data as rates (number of participants enrolled per week across the study period).

The interobserver reliability between paramedics and emergency physicians' use of the CCHR will be calculated using Cohen's kappa, a statistical method for measuring agreement between two observers, together with 95% CIs. Additionally, we will report upon the CCHR's diagnostic performance for ruling out clinically significant TBI. This will be achieved by calculating the sensitivity, specificity, positive and negative likelihood ratios with 95% CIs against a reference standard of traumatic intracranial lesion, which will be identified from in-hospital records. Given the nature of this feasibility study, the analyses will be viewed as exploratory, preliminary and will be descriptive in nature. Furthermore, to determine the acceptability of the CCHR, the OADRI average item scores will be calculated, with higher scores indicating greater acceptability.

### Qualitative data

The six-stage thematic analysis approach will be employed to analyse all interviews, as suggested by Braun and Clarke.[29] This involves becoming familiar with the data, generating initial codes, searching for themes, reviewing themes, defining and naming themes and subthemes, and producing the report. We will adhere to the four criteria for trustworthiness established by Lincoln and Guba: credibility, dependability, confirmability and transferability.[30] Each step of the qualitative data collection and analysis will be guided by a qualitative research expert.

**Table 2** Progression criteria to proceed with a definitive trial

| | Contributing data | Green (proceed to a definitive trial) | Amber (proceed to a definitive trial following some changes to the protocol) | Do not proceed to a definitive trial |
|---|---|---|---|---|
| Screening | Training and delegation logs | At least 20 paramedics at participating ambulance hubs trained and on delegation logs | At least 10 paramedics at participating ambulance hubs trained and on delegation logs | Less than 10 paramedics at participating ambulance hubs trained and on delegation logs |
| Recruitment acceptability | Screening & recruitment log | ≥80% of eligible screened patients are consented to take part | ≥60% of eligible screened patients are consented to take part | <60% of eligible screened patients are consented to take part |
| Retention | Screening & recruitment log | Retention of ≥90% of study participants until study completion | Retention of ≥80% of study participants until study completion | Retention of <80% of study participants until study completion |
| Study procedures | Participation data | At least 85% of the outcome data are collected (eg, case report forms) | A minimum of 60% of the outcome data are collected (eg, case report forms) | <60% of the outcome data are collected (eg, case report forms) |
| Adverse events | Participation data | No or very minor adverse events and no participants discontinued the study | Minor or serious adverse events leading to 10% or less participants discontinuing the study | Serious adverse events leading to >10% of participants discontinuing the study |
| Acceptability | Participant interview Paramedics questionnaire and interviews | Responses to OADRI all 'agree' or 'strongly agree'. No major barriers to acceptability/participation identified during interviews with paramedics and patients | Responses to OADRI mainly 'agree' or 'strongly agree'. Some barriers to participation identified during interviews but remediable action identified | Responses to OADRI mainly 'disagree' or 'strongly disagree'. Major barriers to acceptability/participation identified during interviews with no clear remediable action |

OADRI, Ottawa Acceptability of Decision Rules Instrument.

The qualitative data will be managed and analysed using NVivo V.12.

## Progression criteria

This feasibility study will be overseen and governed by the Trial Steering Committee (TSC), which will include the chief investigator, principal investigator, two lay members, topic experts in TBI and one paramedic. The role of the TSC is to provide external oversight and to ensure that the study is conducted in compliance with Good Clinical Practice guidelines.[31 32]

The traffic light approach (green/amber/red), also known as a (stop–amend–go) rule, will be used to determine whether or not to proceed with a full definitive trial.[33] Green (go) indicates that the study meets the preset criteria, and that it would be feasible for a full-powered definitive trial to be conducted without any modifications to the study protocol. Amber (amend) indicates that further feasibility work or modifications to the protocol are required before proceeding with the full trial. Red (stop) indicates that proceeding to a definitive study is not recommended. Taking into account the feasibility outcomes, as well as qualitative interviews with patients and paramedics, a decision will be made regarding whether or not to proceed with the trial or whether to make any amendments to

the study design. The progression criteria are shown in table 2.

## Patient and public involvement

It has become increasingly apparent that involving patients and the public in healthcare research can enhance the development and implementation of clinical studies, the recruitment of participants, and the relevance of study topics and outcomes in relation to their needs and experiences.[34 35] It will be a priority to take into account the perspectives of patients and the public during the design, conduct, reporting and dissemination of the research. To ensure maximum clinical impact, the study has been discussed with patient and public representatives through the National Institute for Health and Care Research Brain Injury MedTech Co-operative. A comprehensive and in-depth discussion was conducted regarding the study procedures, including key aspects such as the consent process and patient-facing materials. Two members have agreed to join the study steering committee, which is expected to meet two or three times during the study period to advise on the progress of the research, and discuss any adverse events, proposed protocol amendments and other matters relating to research governance. Additionally, a lay summary will

be developed in collaboration with the lay members to disseminate the study findings.

## Safety considerations, adverse events and serious adverse events

Given the nature of this study, any risk to participants is very low. It is important to note that this study is observational in nature: a patient's participation in this study will have no effect on the quality of medical care they usually receive in both prehospital and hospital settings. Our study will have no impact on the decision to undergo a CT scan as this will be undertaken at the discretion of the treating clinician, as part of routine clinical care, in accordance with current national guidance. The study will also have no effect on the decision to transport patients to the hospital. There are also small risks associated with data management, but we have carefully considered how data will be handled during the study and we will operate in accordance with current legislation and the standard operating procedures of the sponsor organisation to mitigate that risk. We are not aware of any potential risks this study would pose to the researchers.

Because of the low-risk nature of the study, we do not anticipate any adverse events (including serious adverse events) as a result of participating in this research. However, given the patient group (patients with suspected TBI), we anticipate that many patients will experience expected adverse events, which are related to their initial injury. We will not ask sites to report expected adverse events. Expected serious adverse events include, for example, death due to head injury, neurosurgical procedures for TBI, admission to hospital for traumatic brain injury and loss of function due to TBI. Unexpected adverse events and serious adverse events should be reported. Adverse events will be defined in accordance with ICH (International Council for Harmonisation) Good Clinical Practice as any untoward medical occurrence in a participant, but will only be reported if this was not expected as a result of the patient's condition, up to the point of handover in hospital.

## Impact

It is anticipated that the feasibility study will provide valuable insight into whether conducting a full trial in the future is warranted. It will also provide guidance to enhance the study design, recruitment plan and consent process. If feasibility is established, we will prepare to run a fully powered multicentre study to evaluate the diagnostic accuracy of using the CCHR in the prehospital field. This will also involve drafting a proposal to apply for research funding.

The outcomes relating to diagnostic accuracy will be of particular relevance given our knowledge that this is the first study investigating the feasibility of using the CCHR in the prehospital environment. Therefore, it is anticipated that the findings of this feasibility study will be used to inform design of a fully powered diagnostic test accuracy study. The findings of this study will be complemented by the findings of mixed-methods research that we are also conducting which will explore facilitators and barriers to implementing new diagnostic pathways in the prehospital environment as perceived by practising paramedics.[36]

In parallel with a fully powered diagnostic test accuracy study, it will also be important to work with key stakeholders including primary care practitioners, neurosurgeons and patient and public representatives. The care pathway for patients who are not conveyed to hospital will need to be defined to ensure that patients are adequately supported with appropriate safety netting for those who experience ongoing symptoms or complications.

## ETHICS AND DISSEMINATION

Ethical approval to conduct this feasibility study has been obtained from the Health Research Authority and the Research Ethics Committee (REC reference: 22/NW/0358). Any substantial amendment to the study protocol will need to be approved by an ethics committee after being approved by the sponsor. To ensure privacy and confidentiality, all records and files related to the study will be kept in a secure location and will only be accessible to members of the research team. The results will be published in peer-reviewed journals, presented at relevant national and international conferences, and will be incorporated into a doctoral thesis (NA). The study findings will also be disseminated through presentations at relevant academic forums.

**Author affiliations**
[1]Division of Cardiovascular Sciences, The University of Manchester, Manchester, UK
[2]Department of Accidents and Trauma, Prince Sultan bin Abdelaziz College for Emergency Medical Services, King Saud University, Riyadh, Saudi Arabia
[3]Medical Directorate, North West Ambulance Service NHS Trust, Bolton, UK
[4]Emergency Department, Manchester University NHS Foundation Trust, Manchester, UK
[5]School of Health and Related Research, University of Sheffield, Sheffield, UK

**Contributors** NA collaborated with the study team (SB, SDC, FL and RB) in preparing the initial study protocol. The protocol was developed through collective input and discussions. NA took the lead in drafting the manuscript, working closely with SB, SDC, FL and RB to ensure its accuracy and completeness. The final version of the manuscript was approved by all authors.

**Funding** This work is supported by a PhD scholarship awarded to NA by the Saudi Arabian Cultural Bureau in the UK and King Saud University (KSU1872). The study is sponsored by the University of Manchester. The publication fee is funded by the Cardiovascular Department, University of Manchester, UK (no grant or award number).

**Disclaimer** The funders had no role in the design of this study protocol.

**Competing interests** RB has consulted for Abbott Point of Care and is undertaking contract clinical research with BrainBox, which manufactures assays for biomarkers of TBI.

**Patient and public involvement** Patients and/or the public were involved in the design, or conduct, or reporting, or dissemination plans of this research. Refer to the Methods section for further details.

**Patient consent for publication** Not applicable.

**Provenance and peer review** Not commissioned; externally peer reviewed.

peer-reviewed. Any opinions or recommendations discussed are solely those of the author(s) and are not endorsed by BMJ. BMJ disclaims all liability and responsibility arising from any reliance placed on the content. Where the content includes any translated material, BMJ does not warrant the accuracy and reliability of the translations (including but not limited to local regulations, clinical guidelines, terminology, drug names and drug dosages), and is not responsible for any error and/or omissions arising from translation and adaptation or otherwise.

**ORCID iDs**
Naif Alqurashi http://orcid.org/0000-0003-4417-4128
Steve Bell http://orcid.org/0000-0003-3891-6623
Fiona Lecky http://orcid.org/0000-0001-6806-0921

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
