## [Reviewer comments · BMJ Open]

ARTICLE DETAILS

TITLE (PROVISIONAL)	Head Injury Evaluation and Ambulance Diagnosis (HOME) Study Protocol: A Feasibility Study Assessing the Implementation of the Canadian Computed Tomography Head Rule in the Prehospital Setting
AUTHORS	Alqurashi, Naif; Bell, Steve; Carley, Simon D; Lecky, Fiona; Body, Richard

VERSION 1 – REVIEW

REVIEWER	Papa, Linda Orlando Regional Medical Center, Emergency Medicine
REVIEW RETURNED	03-Oct-2023

GENERAL COMMENTS	This seems like a very straight forward study. CCHR is a very valuable clinical tool. The CCHR was created to determine the need for CT scan. There needs to be a more convincing rationale for why a paramedic or EMT should be making this decision. Please describe in more detail how this will improve practice. A sample size of 100 is low (even for a feasibility study) given that very few patients will have a positive CT. A power calculation should be conducted to ensure that enough positive CTs will be included to make a conclusion about feasibility and inter-rater reliability. Having data captured both on paper and electronically will be messy. Paper can easily be misplaced and valuable data lost. How long will the paramedic have to complete the data collection form? It is unclear if paramedics will have time to complete the data enroute. There could be an opportunity for data contamination once the paramedic speaks to the treating ED physician. How will this be handled? If they wait too long to complete the form, there is the chance of introducing recall bias. How will missing data be handled? The exact (location, type, time) of training for the involved paramedics is not provided.
---

	Please describe the prehospital system in which this will be studied (how similar/different is to other systems). Will there be quality assurance for paramedics to ensure they apply the decision rule uniformly? Please describe how and how often this will occur. Will any decision rule elements be extracted from the medical record? Why is the plan to ensure that forms are completed prior to the CT. How long will consent be deferred? The statistical analysis section is very broad. There is a lack of description of the exact data that will be used to ultimately determine feasibility i.e., how will the go-no-go criteria be handled statistically.
--	--

REVIEWER	Carver, Thomas Medical College of Wisconsin, Surgery
REVIEW RETURNED	04-Nov-2023

GENERAL COMMENTS	This study seems well-designed although I am still a little unclear on the definition of head injury. Isolated head injury from a fall or from an assault is one thing, a motor vehicle crash is another, and I really cannot tell if you are excluding patients in an MVC. If you are, why? You could easily evaluate this tool even in patients who have to be evaluated at an MTC because of their MVC and I think it will give you meaningful information. Just a thought.
--

VERSION 1 – AUTHOR RESPONSE

Reviewer: 1:

C1: This seems like a very straight forward study. CCHR is a very valuable clinical tool. The CCHR was created to determine the need for CT scan. There needs to be a more convincing rationale for why a paramedic or EMT should be making this decision. Please describe in more detail how this will improve practice.

R1: Thank you for your valuable comments. In the background section of the manuscript, I have provided a comprehensive explanation of how incorporating the CCHR into prehospital care can enhance practice.

C2: A sample size of 100 is low (even for a feasibility study) given that very few patients will have a positive CT. A power calculation should be conducted to ensure that enough positive CTs will be included to make a conclusion about feasibility and inter-rater reliability.

R2: Thank you so much. Regarding your concern about the sample size of 100 being low for our study, we have very carefully considered this point. In guidance for the design and conduct of feasibility studies, Teresi et al. (2022) state that "the sample size should be based on "practical considerations including participant flow, budgetary constraints, and the number of participants

needed to reasonably evaluate feasibility goals." For qualitative work, to reach saturation, sample sizes may be 30 or less. For quantitative studies, a sample of 30 per group (intervention and control) may be adequate to establish feasibility."

<https://www.ncbi.nlm.nih.gov/pmc/articles/PMC8849521/#S7>

Further, the primary aim of this feasibility study is not to definitively determine the diagnostic accuracy of CCHR but to assess whether a larger, fully-powered study would be feasible in terms of logistics, participant recruitment, and methodology. Although we will calculate sensitivity (as an early estimate), we anticipate that the 95% CI will be too wide to provide a generalisable impression of the sensitivity of the CCHR. To do so will require a much larger sample size and can therefore only be meaningfully evaluated in a much larger definitive study. The insights we will gain from this study will inform the design of the larger-scale research study. I have added a sentence to clarify this with a reference - (see Study duration and sample size section).

C3: Having data captured both on paper and electronically will be messy. Paper can easily be misplaced, and valuable data lost.

R3: Our paper forms are not the primary means of data collection. Paramedics will be asked to use electronic data collection via REDCap. However, recognising the unpredictable nature of fieldwork, we will maintain paper forms as a backup plan.

C4: How long will the paramedic have to complete the data collection form? It is unclear if paramedics will have time to complete the data enroute.

R4: We have piloted the REDCap data collection form and determined that it takes paramedics approximately 3-4 minutes to complete. This duration is deemed feasible and practical for their workflow. Our team, which includes paramedics with extensive field experience, has also confirmed its feasibility. We agree with the reviewer. However, it is important to note that a fundamental aspect of our feasibility study is to assess whether this approach is practical in real-world practice.

C5: There could be an opportunity for data contamination once the paramedic speaks to the treating ED physician. How will this be handled?

R5: We will ask paramedics not to hand over the results of the Canadian CT head rule assessment. Of note, paramedics typically hand over patients to triage nurses, not directly to ED physicians, except in cases where a major trauma team is activated.

C6: If they wait too long to complete the form, there is the chance of introducing recall bias.

R6: Paramedics will be encouraged to complete the REDCap form as soon as possible, ideally before or immediately after patient handover.

C7: How will missing data be handled?

R7: Evaluating the proportion of missing data is one of the objectives. They won't be imputed because quantifying missing data is an objective in itself.

C8: The exact (location, type, time) of training for the involved paramedics is not provided.

R8: We have created a comprehensive training program that accommodates their busy schedules. "Prior to their participation in the study, all participating paramedics will be provided with bespoke training specifically designed for the study. This training will cover all aspects of the study protocol, including eligibility assessment, the consent process, data collection procedures, and guidance on interpreting the CCHR. To ensure accessibility and flexibility, the training will be offered both face-to-face and through various online sessions conducted via Zoom. Additionally, the training sessions have been recorded and made available online for those unable to attend live. This approach allows paramedics to access the training material at their convenience, ensuring that all participants receive comprehensive training on the study protocols and data collection methods." I added a couple of sentences in the manuscript to cover these aspects (see "Data collection section").

C9: Please describe the prehospital system in which this will be studied (how similar/different is to other systems).

R9: I have included information about the North West Ambulance Service in the 'study design and setting' section of the manuscript.

C10: Will there be quality assurance for paramedics to ensure they apply the decision rule uniformly? Please describe how and how often this will occur.

R10: Yes, quality assurance is a key component of our study to ensure that paramedics apply the Canadian CT Head Rule uniformly. To achieve this, we have developed a comprehensive training program that all participating paramedics will undergo prior to the study. This training, delivered both face-to-face and through online sessions, covers detailed instructions on the application of the decision rule. Further, assessing interobserver reliability is a primary objective of our study. However, this assessment will not be limited to paramedics alone, but it will also involve comparing their application of the rule with that of treating doctors.

C11: Will any decision rule elements be extracted from the medical record?

R11: Our study design does not include this process.

C12: Why is the plan to ensure that forms are completed prior to the CT.

R12: This is essential because, after the CT, the doctor will be aware of the patient's outcome, which could introduce bias.

C13: How long will consent be deferred?

R13: We did not specify this in the protocol that was submitted for ethical approval, nor in our application form for ethical approval. In practice, the first approach is likely to be made shortly after the potential participant arrives in hospital unless there is ongoing reason to suspect that the participant's mental capacity to consent is compromised. If so, the potential participants will be reviewed again while in hospital (where applicable and feasible) or they will be contacted after hospital discharge. No time limit has been specified in the approved version of the protocol.

C14: The statistical analysis section is very broad. There is a lack of description of the exact data that will be used to ultimately determine feasibility i.e., how will the go-no-go criteria be handled statistically.

R14: We have expanded upon the 'Data Analysis' section of the manuscript to specifically detail the data that will be used to determine feasibility. The stop/amend/go progression criteria will be handled statistically by setting specific thresholds for various metrics that indicate whether the study should proceed, require modifications, or should not proceed to a definitive trial. This has been clarified in Table 2.

Reviewer: 2:

C1: This study seems well-designed although I am still a little unclear on the definition of head injury. Isolated head injury from a fall or from an assault is one thing, a motor vehicle crash is another, and I really cannot tell if you are excluding patients in an MVC. If you are, why? You could easily evaluate this tool even in patients who have to be evaluated at an MTC because of their MVC and I think it will give you meaningful information.

R2: Thank you so much for reviewing our study protocol. I would like to clarify that MVC patients will not be excluded. However, we will exclude cases where there is evidence of multisystem trauma as assessed by the treating paramedic. This is important because our focus is on evaluating the diagnostic tool in cases involving head injuries, rather than multisystem trauma.

VERSION 2 – REVIEW

REVIEWER	Papa, Linda Orlando Regional Medical Center, Emergency Medicine
REVIEW RETURNED	22-Dec-2023

GENERAL COMMENTS	Thank you to the authors for their responses. Please clarify the following items. The rationale for why a paramedic or EMT should be making this decision needs stronger support. There are no references to justify that incorporating CCHR in the field will: 1) reduce healthcare costs, 2) improve practices, 3) reduce response times or 4) more efficiently identify patients with less serious head injuries. There has to be stronger supporting evidence to justify having this decision made in the field and to ensure it would not compromise patient safety. Furthermore, just because a CT is negative does not mean the patient does not need medical evaluation for a concussion. What about denying patients with mild TBI appropriate medical care? Please address. In this system (receiving hospitals), how many CTs are performed on mild TBI patients (50%, 70%, 90%)? Is the CCHR used regularly in these hospitals?
---

	What are the exact criteria for excluding patients with multiple trauma? These have to be precisely defined so all paramedics apply them uniformly. Interpretation of the CCHR can vary among ED physicians. How will this variability be assessed among paramedics? Although a total of 100 seems reasonable, the problem is the number of positive CTs. The sample size of 100 is too low because there will be an inadequate number of CT positive cases. There will likely be less than 5 CT positive cases. A power calculation should be conducted. The power for a feasibility study is lower than a full-scale study but it needs to be performed, nonetheless. How will investigators determine that EMTs will consistently administer the CCHR? Is there an exam to test their knowledge? The question of deferred consent is critical in human subjects research. This has to be defined in advance. It is still not clear if paper is going to be required for all cases as a back-up. What is the defined period for paramedics to absolutely complete the data collection form (24 hours, 72 hours)? Where will the money come from for the vouchers? What kind of funding? Please precisely define minor adverse events. Statistically weigh the “amber” items of the go-no-go criteria.
--	---

VERSION 2 – AUTHOR RESPONSE

Response to Reviewers’ comments:

We would like to thank the reviewer and editorial staff for their constructive and insightful comments. We have addressed them, as detailed below, and we hope you will agree that this has improved the manuscript.

C1: The rationale for why a paramedic or EMT should be making this decision needs stronger support. There are no references to justify that incorporating CCHR in the field will: 1) reduce healthcare costs, 2) improve practices, 3) reduce response times or 4) more efficiently identify patients with less serious head injuries. There has to be stronger supporting evidence to justify having this decision made in the field and to ensure it would not compromise patient safety.

R1: In the introduction, we provided clarification to better justify the use of the CCHR by prehospital care providers. We highlighted the high volume of minor head injury cases and referenced the successful use of other clinical decision tools, such as the HEART score for chest pain management, to reduce unnecessary hospital visits and costs. This serves as an illustration of the potential benefits clinical decision tools can bring to prehospital care.

Direct evidence from prehospital application of the CCHR is currently lacking due to the fact that this tool has not been studied in the prehospital field. This gap highlights the innovative aspect of our study and highlights the importance of this feasibility study.

C2: Furthermore, just because a CT is negative does not mean the patient does not need medical evaluation for a concussion. What about denying patients with mild TBI appropriate medical care? Please address.

R2: In the first step here is to assess the accuracy of the CCHR. Once we establish its accuracy through the larger study, we plan to consider an interventional trial that might involve leaving patients at the scene. This future phase will require careful planning to ensure that even patients who are not transported receive appropriate follow-up care, particularly for conditions such as post-concussion syndrome, thereby ensuring that important patients are not missed.

Our goal is to enhance prehospital TBI care without compromising patient safety or the comprehensive assessment and management of TBI, ensuring no significant cases are missed. To address the important point you have made, we have added the following text under 'impact' to acknowledge that additional work will also need to be undertaken to define the care pathway for patients who are not conveyed:

“In parallel with a fully powered diagnostic test accuracy study, it will also be important to work with key stakeholders including primary care practitioners, neurosurgeons and patient and public representatives. The care pathway for patients who are not conveyed to hospital will need to be defined to ensure that patients are adequately supported with appropriate safety netting for those who experience ongoing symptoms or complications”.

C3: In this system (receiving hospitals), how many CTs are performed on mild TBI patients (50%, 70%, 90%)? Is the CCHR used regularly in these hospitals?

R3: We do not currently have the exact percentages available for our manuscript. However, we can confirm that the CCHR is employed in a modified form at these hospitals, in alignment with the NICE guidelines.

C4: What are the exact criteria for excluding patients with multiple trauma? These have to be precisely defined so all paramedics apply them uniformly.

R4: Here, we have decided to adopt a pragmatic approach that relies on the clinical judgment of the treating paramedics, defining multisystem trauma as the involvement of more than one bodily system. Therefore, we have defined the exclusion criterion based on a paramedic's professional judgment of multisystem trauma, where the involvement of more than one bodily system indicates exclusion from the study. When paramedics assess patients and identify injuries that extend beyond isolated head trauma, such patients will be excluded. We have now included this into the manuscript.

C5: Interpretation of the CCHR can vary among ED physicians. How will this variability be assessed among paramedics?

R5: This is an important point. Our protocol addresses this through the assessment of inter-observer reliability between paramedics and ED clinicians, as detailed in the methods section. We strongly considered evaluating the inter-observer reliability between paramedics. However, this would require two paramedics to attend patients, which does not reflect current practice and would be unfeasible given current service pressures. The absence of this assessment should not affect our decision about the feasibility of a full study.

C6: Although a total of 100 seems reasonable, the problem is the number of positive CTs. The sample size of 100 is too low because there will be an inadequate number of CT positive cases. There will likely be less than 5 CT positive cases. A power calculation should be conducted. The power for a feasibility study is lower than a full-scale study but it needs to be performed, nonetheless.

R6: We appreciate the reviewer's concern regarding the sample size and the anticipated number of positive CT cases in our feasibility study. It's important to clarify that the primary aim of this phase is to evaluate trial processes, such as the feasibility of the consent procedure, data collection methods, and the practical implementation of the CCHR in a prehospital setting, rather than to achieve statistical power for clinical outcomes.

Following the guidance for the design and conduct of feasibility studies by Teresi et al. (2022), “the sample size should be based on practical considerations, including participant flow, budgetary constraints, and the number of participants needed to reasonably evaluate feasibility goals. For qualitative work, sample sizes may be 30 or less to reach saturation. For quantitative studies, a sample of 30 per group (intervention and control) may be adequate to establish feasibility.” This is a feasibility study aimed at examining trial processes, including the feasibility of consent and the data collection process, among others. We think that an extensive sample size is not required at this stage. This will be a consideration for the main study. The main study will be appropriately powered to include an adequate number of positive CT cases. The insights gained from this feasibility study will inform the design of the larger-scale research study.

<https://www.ncbi.nlm.nih.gov/pmc/articles/PMC8849521/#S7>

C7: How will investigators determine that EMTs will consistently administer the CCHR? Is there an exam to test their knowledge?

R7: We appreciate the opportunity to further clarify the measures in place to ensure the consistent administration of the CCHR by participating paramedics. As stated in our manuscript, comprehensive training was provided both face-to-face and through online sessions, with recorded materials available for flexible access. This approach ensures that all participating paramedics receive comprehensive training on the study protocol and data collection methods related to the CCHR.

We will also assess inter-observer reliability of the assessment between paramedics and ED clinicians, which will ensure consistency with the current standard of care.

Furthermore, we maintain that the CCHR, due to its straightforward criteria and the comprehensive training provided to paramedics, does not require this additional layer of complexity. We believe that this decision aligns with our primary objective to examine the feasibility of using the CCHR in the prehospital field.

C8: The question of deferred consent is critical in human subjects research. This has to be defined in advance.

R8: We would like to clarify that we have established a comprehensive protocol for deferred consent prior to the patient enrolment. This protocol is detailed as follows:

- Paramedics will enrol patients given that immediate informed consent is not feasible due to the nature of the injury and potential impairment of the patients' mental capacity.
- Deferred consent will be sought as soon as possible after the patients have received initial treatment and are considered to have regained sufficient mental capacity to provide informed consent (as soon as feasible post-initial treatment).
- If a participant does not regain capacity, we will seek consent from their legally authorised representative or a family member at the earliest opportunity.

In the manuscript, we have already described how we will approach deferred consent and ensure compliance with ethical standards. We will also ensure that all participants or their representatives are fully informed about the study, their involvement, and their rights, including the right to withdraw, through a detailed participant information sheet provided at the time of seeking deferred consent. This approach has been approved by the Sponsor organisation, the NHS Research Ethics Committee and the Health Research Authority (HRA).

We have now added a sentence to define the deferred consent process.

C9: It is still not clear if paper is going to be required for all cases as a back-up.

R9: We would like to clarify that the paper CRFs are a backup method for data collection, specifically in situations where the iPad is not functioning or when access to the REDCap system is temporarily unavailable. This ensures that the data collection process can continue without interruption.

C10: What is the defined period for paramedics to absolutely complete the data collection form (24 hours, 72 hours)?

R10: Paramedics are instructed and encouraged to complete the forms promptly upon arriving at the hospital, ideally before or immediately following patient handover, to ensure accurate and immediate data capture. We have added the following sentence to the manuscript (p. 14):

“Paramedics will record these data contemporaneously, either prior to or immediately following handover at hospital”.

The protocol does not allow for retrospective data entry.

C11: Where will the money come from for the vouchers? What kind of funding?

R11: We would like to clarify that the funds for the vouchers will be derived from the bench fees allocated for the PhD candidate, Naif Alqurashi. We have secured the necessary funds for this project.

C12: Please precisely define minor adverse events.

R12: We have revised the relevant section (Safety considerations, adverse events and serious adverse events) of our manuscript to include a clear definition of minor adverse events. We have added the following text:

“Adverse events will be defined in accordance with ICH Good Clinical Practice as any untoward medical occurrence in a participant, but will only be reported if this was not expected as a result of the patient’s condition, up to the point of handover in hospital”.

C13: Statistically weigh the “amber” items of the go-no-go criteria.

R13: We have reviewed the contents of the text and Table 2, and it appears that we have quantified the amber criteria. The criteria are indicative and ultimately the Trial Steering Committee will make the decision about feasibility, taking account of both quantitative and qualitative data emerging from the study.

VERSION 3 – REVIEW

REVIEWER	Papa, Linda Orlando Regional Medical Center, Emergency Medicine
REVIEW RETURNED	05-Apr-2024

GENERAL COMMENTS	Again, thank you to the authors for responding to the comments. Please remove the reference to reduced costs from the heart score and focus on costs for TBI instead. In the introduction, please state how healthcare costs have been impacted by the use of CCHR in the emergency department. How many CTs have been reduced? How has practice improved? Please provide concrete examples and references. One of the high-risk criteria for the CCHR is “GCS less than 15 at 2 hours.” This cannot be captured by paramedics. If a patient worsens after being told they do not have to go to the hospital (e.g. epidural hematoma), that is a significant adverse event. How will this very important safety issue be addressed? Given that patients with GCS 13 and 14 are at much higher risk of intracranial lesions would it be wise to exclude them? For research purposes, exclusion criteria for patients multiple trauma need to be defined. This provides guidance, so all paramedics apply them uniformly. Leaving this entirely open to paramedics introduces great variability and bias. At least 30% of patients with mTBI continue to have problems after injury. Please elaborate how follow-up resources are going to be provided to patients.
---

	In this system (receiving hospitals), how many CTs are performed on mild TBI patients (50%, 70%, 90%)? Given these numbers, how many additional CT scans would be saved. Sample size is based on the outcome selected for a particular study and should not be generalized. In this feasibility study, it is crucial that an adequate number of cases with positive CTs are included. The power for a feasibility study is lower than a full-scale study but it needs to be performed, nonetheless.
--	---

VERSION 3 – AUTHOR RESPONSE

Response to Reviewer’s comments:

C1: Please remove the reference to reduced costs from the heart score and focus on costs for TBI instead. In the introduction, please state how healthcare costs have been impacted by the use of CCHR in the emergency department. How many CTs have been reduced? How has practice improved? Please provide concrete examples and references.

R1: We have revised the introduction section of our manuscript accordingly. We now emphasise the role of NICE guidelines (UK guidelines), which largely follow the CCHR in determining the necessity of CT scans for head injury cases. We have incorporated specific evidence demonstrating the cost-effectiveness of the CCHR, supported by studies showing a reduction in hospital admission rates.

Direct evidence from prehospital application of the CCHR is currently lacking due to the fact that this tool has not been studied in the prehospital field. This gap highlights the innovative aspect of our study and highlights the importance of this feasibility study.

As part of our efforts to adhere to word count limits, we also edited the first and third paragraphs, taking care to maintain the essential meanings of each paragraph.

C2: One of the high-risk criteria for the CCHR is “GCS less than 15 at 2 hours.” This cannot be captured by paramedics. If a patient worsens after being told they do not have to go to the hospital (e.g. epidural hematoma), that is a significant adverse event. How will this very important safety issue be addressed?

R2: As noted in our initial submission, we recognised the limitations paramedics face in assessing the Glasgow Coma Scale score at 2 hours post-injury. To address this, we amended/modified the CCHR criteria to evaluate the GCS score at the time of initial contact (patient assessment), which was clearly stated in our manuscript under Table 1. In the proposed diagnostic test accuracy study that we are evaluating the feasibility of, we will establish whether this modified tool can be used while preserving

the accuracy that has been demonstrated with the original tool when applied in Emergency Departments.

C3: Given that patients with GCS 13 and 14 are at much higher risk of intracranial lesions would it be wise to exclude them?

R3: Those patients will be identified as high risk by the CCHR. It is important to retain them in the analysis because the CCHR is intended to be applied to all these patients. Our evaluation will therefore provide a real-world evaluation of diagnostic test accuracy. If we exclude patients who will test positive with the rule, we may artificially inflate specificity and the 95% CI for sensitivity will be extremely wide, even with a very large study, due to the lower prevalence of TBI. Therefore, we opted to retain these participants, which is how the CCHR was intended to be applied in practice.

C4: For research purposes, exclusion criteria for patients multiple trauma need to be defined. This provides guidance, so all paramedics apply them uniformly. Leaving this entirely open to paramedics introduces great variability and bias.

R4: We understand the importance of uniform application of these criteria to minimise variability and potential bias in our study.

In our manuscript, we have previously specified the exclusion criteria as follows: patients with trauma to other body regions requiring clinical treatment, which indicates multisystem trauma, are excluded. This typically includes patients involved in MVCs where the treating paramedic assesses the presence of injuries beyond isolated head trauma that require clinical intervention.

We believe that the criteria for identifying multisystem trauma are clearly outlined in the study protocol. All paramedics involved in the study undergo specific training sessions focused on the study protocol and these criteria. These sessions include detailed explanations of the inclusion and exclusion criteria.

Also, we have developed an assessment checklist to be used by paramedics at the scene as part of assessing and confirming the eligibility. This checklist will guide the paramedics through a step-by-step assessment process to determine if a patient meets any of the exclusion criteria. This was designed to minimise subjective interpretation and enhance consistency in patient assessment.

C5: At least 30% of patients with mTBI continue to have problems after injury. Please elaborate how follow-up resources are going to be provided to patients.

R5: In this study, the care of participants will not be changed. All participants will be transported to hospital and will receive usual care. The treating clinicians will determine the need for ongoing follow up for persistent symptoms. If we demonstrate that the CCHR has adequate diagnostic accuracy for use in practice in our definitive study, we will need to work with stakeholders to establish a system that will enable patients who are left at the scene to access the same follow-up resources. This will therefore be a goal for future work.

C6: In this system (receiving hospitals), how many CTs are performed on mild TBI patients (50%, 70%, 90%)? Given these numbers, how many additional CT scans would be saved.

R6: Thank you for your continued interest in the specifics of CT scan usage for mild TBI patients within the receiving hospitals. As mentioned in our previous response, we currently do not have access to specific data regarding the exact percentage of CT scans performed on mild TBI patients in these facilities. This information is not publicly available, gathering it is not incorporated in our ethically approval and therefore it has not been possible to obtain these data from the hospitals involved in this study. However, we do not feel that this will affect the conduct of the study in any way.

C7: Sample size is based on the outcome selected for a particular study and should not be generalized. In this feasibility study, it is crucial that an adequate number of cases with positive CTs are included. The power for a feasibility study is lower than a full-scale study but it needs to be performed, nonetheless.

R7: We thank the reviewer for continuing to emphasise the importance of the sample size for our feasibility study. As previously addressed, our approach is guided by standard practices in feasibility research, focusing on evaluating logistical and procedural aspects rather than achieving definitive clinical outcomes.

The primary objective of this feasibility study is not to conclusively determine the diagnostic accuracy of the CCHR but to assess and refine the logistics, participant recruitment, and application of the CCHR in the prehospital setting. The expected number of positive CT cases, although likely to be limited, will still provide valuable preliminary data on the application of the CCHR. Clearly, that is insufficient to determine sensitivity. However, the sample size will be sufficient to establish the feasibility of study processes, which is our objective. We will proceed to evaluate the CCHR in the larger study if we establish the feasibility of such a study. This larger study will be appropriately powered to statistically validate clinical outcomes, including a more robust analysis of sensitivity and specificity. This is also typically acceptable in feasibility studies where the primary goal is not to test hypotheses definitively but to prepare for larger-scale studies.

In line with feasibility study practices, we are focusing on ensuring that the study design, recruitment processes, and data collection methods are robust and adaptable. These steps are essential to ensure the success and validity of the future definitive study. Therefore, while we acknowledge the concerns about the small number of positive CT cases in this initial phase, our approach is strategically planned to balance these limitations with the practical considerations and objectives of a feasibility study. In addition, this sample size is considered sufficient to identify any significant logistical issues, to evaluate the training needs of prehospital personnel, and to assess the initial acceptability of the CCHR application in this setting.

Feasibility studies do not typically support robust statistical conclusions about clinical hypotheses but should identify major operational issues and gather preliminary data. A sample of 100 patients allows for meaningful feedback on these operational and logistical aspects and provides a solid basis for decisions about proceeding to a full-scale trial.

Our approach aligns with established research practices, influenced by recent guidelines and literature on pilot and feasibility studies. With this sample size, we believe that we can reliably evaluate the practicality of using the CCHR in prehospital settings and make informed decisions about progressing to a full-scale study.

Given the objectives of feasibility studies, our chosen sample size is appropriate for assessing the practical implementation aspects of the CCHR in the prehospital setting, thereby preparing for a well-designed definitive trial. We appreciate the opportunity to clarify these points further and believe our approach is well-justified within the context of feasibility research.

References used to write this repones

- Lewis M, Bromley K, Sutton CJ, McCray G, Myers HL, Lancaster GA. Determining sample size for progression criteria for pragmatic pilot RCTs: the hypothesis test strikes back!. *Pilot and feasibility studies*. 2021 Dec;7:1-4.
- Teresi JA, Yu X, Stewart AL, Hays RD. Guidelines for designing and evaluating feasibility pilot studies. *Medical care*. 2022 Jan 1;60(1):95-103.
- Totton N, Lin J, Julious S, Chowdhury M, Brand A. A review of sample sizes for UK pilot and feasibility studies on the ISRCTN registry from 2013 to 2020. *Pilot and Feasibility Studies*. 2023 Nov 21;9(1):188.

VERSION 4 – REVIEW

REVIEWER	Papa, Linda Orlando Regional Medical Center, Emergency Medicine
REVIEW RETURNED	05-May-2024

GENERAL COMMENTS	In the protocol use the number of positive CTs as an endpoint. For instance, when there are 20 positive CT scans, that can be your endpoint (rather than a total number of patients). One of the high-risk criteria for the CCHR is “GCS less than 15 at 2 hours.” If a patient worsens after being told they do not have to go to the hospital (e.g. epidural hematoma), that is a significant adverse event. Will the paramedics advise patients to go to the hospital should their mental status worsen? Will they recommend that they should have someone check on them (neuro checks)?
---

VERSION 4 – AUTHOR RESPONSE

Response to Reviewer’s comments:

C1: In the protocol use the number of positive CTs as an endpoint. For instance, when there are 20 positive CT scans, that can be your endpoint (rather than a total number of patients).

R1: We have added the number and proportion of participants with neurosurgically significant TBI as an outcome (see the ‘Outcomes’ section in the manuscript). We unfortunately cannot commit to a minimum sample size of 20 participants with neurosurgically important TBI given funding and time constraints associated with this feasibility study. (This is likely to require recruitment of over 400 participants in total, and failure to do so wouldn’t imply lack of feasibility but rather the funding and time constraints that we are operating under). However, it will allow us to establish prevalence, which will inform the sample size for the definitive study. Our approach ensures that we gather essential data to inform the feasibility and design of a larger-scale study. This strategy aligns with best practices in feasibility study design, where the primary focus is on testing logistical and procedural feasibility rather than on achieving specific clinical endpoints.

C2: One of the high-risk criteria for the CCHR is “GCS less than 15 at 2 hours.” If a patient worsens after being told they do not have to go to the hospital (e.g. epidural hematoma), that is a significant adverse event. Will the paramedics advise patients to go to the hospital should their mental status worsen? Will they recommend that they should have someone check on them (neuro checks)?

R2: In this feasibility study, the care of participants will not be changed. All patients will be transported to hospital and will receive the usual care. Therefore, these participants will still be transported to hospital. Such events will be captured and classified as neurosurgically significant TBI. This will then be evident in the evaluation of the diagnostic accuracy of the CCHR, which is an objective of the definitive study. During this observational study, no participants are at risk of under-diagnosis. However, the study will establish whether implementing the CCHR in the prehospital environment would lead to under-diagnosis.